# Tight Robustness Certificates and Wasserstein Distributional Attacks for Deep Neural Networks

## Abstract

Wasserstein distributionally robust optimization (WDRO) provides a framework for adversarial robustness, yet existing methods based on global Lipschitz continuity or strong duality often yield loose upper bounds or require prohibitive computation. In this work, we address these limitations by introducing a primal approach and adopting a notion of *exact* Lipschitz certificate to tighten this upper bound of WDRO. In addition, we propose a novel Wasserstein distributional attack (WDA) that directly constructs a candidate for the worst-case distribution. Compared to existing point-wise attack and its variants, our WDA offers greater flexibility in the number and location of attack points. In particular, by leveraging the piecewise-affine structure of ReLU networks on their activation cells, our approach results in an *exact* tractable characterization of the corresponding WDRO problem. Extensive evaluations demonstrate that our method achieves competitive robust accuracy against state-of-the-art baselines while offering tighter certificates than existing methods.

## 1 Introduction

Modern deep networks achieve remarkable accuracy yet remain fragile to distribution shift and adversarial perturbations (Szegedy et al., 2014; Goodfellow et al., 2014; Kurakin et al., 2018; Hendrycks & Dietterich, 2019; Ovadia et al., 2019; Taori et al., 2020; Koh et al., 2021), raising concerns about their reliability in deployment. A principled avenue for robustness is Wasserstein distributionally robust optimization (WDRO, Mohajerin Esfahani & Kuhn 2018; Gao & Kleywegt 2023), which controls worst-case test risk over an ambiguity set within a Wasserstein ball around the empirical distribution and admits tight dual characterizations from optimal transport (Villani et al., 2008; Santambrogio, 2015). While numerous defenses have been proposed, a fundamental gap persists between theoretical robustness certificates and practical adversarial evaluation: existing Lipschitz-based certificates often provide loose upper bounds that vastly overestimate the true worst-case loss (Virmaux & Scaman, 2018), while standard attacks restrict perturbations to fixed-radius balls around individual points (Katz et al., 2017; Ehlers, 2017; Weng et al., 2018; Singh et al., 2018). This mismatch stems from two limitations: certificates typically rely on global worst-case analysis that ignores the actual network geometry traversed by data, and attacks consider only point-wise perturbations rather than distributional shifts permitted by Wasserstein threat models (Singh et al., 2018; Gao & Kleywegt, 2023). The discrepancy is particularly pronounced for modern architectures with ReLU activations, where the network behaves as a piecewise-affine function whose local properties vary dramatically across regions (Jordan & Dimakis, 2020), and those with smooth activations (GELU, SiLU/Swish) exhibit complex nonlinear geometry (Hendrycks & Gimpel, 2016; Ramachandran et al., 2017; Elfwing et al., 2018). In this work, we aim to address both sides of this gap: our contributions can be summarized as follows.

1. For a class of networks with Rectified Linear Unit (ReLU) activations (Nair & Hinton, 2010), we analyze the upper and lower bounds of the Wasserstein Distributional Robust Optimization (WDRO) problem by connecting with the tight Lipschitz constant studied in Jordan & Dimakis, 2020. Our analysis is based on the classical underlying piecewise-affine structure of ReLU networks: on any strict ReLU cell, the logit map $\theta(\cdot)$ is affine with a constant input-logit Ja-

cobian $J_D$. Our contribution is to leverage this structure for WDRO, which requires combining the Lipschitz constant of the logit map and the sensitivity of the softmax cross-entropy, or the DLR loss. Our first theoretical result yields an upper bound of WDRO induced by $L \triangleq 2^{1/s} \max_{D \in \mathcal{D}_{\mathcal{X}}} \|J_D\|_{r \to s}$, where $J_D$ is general Jacobian of the logit map. (See 3.1 for precise definition of $J_D$.) In addition, we derive a lower bound of WDRO by constructing a concrete and finite worst-case distribution. (See equation 16 for the explicit formulation.) This worst-case distribution is constructed by perturbing the empirical sample along the direction in which the logit map is most varied. Moreover, we provide a sufficient condition where our lower and upper bounds match, and simulate an instance to illustrate this tightness, see Figure 2a.

2. We further analyze the upper and lower bounds of the Wasserstein Distributional Robust Optimization (WDRO) problem for a class of MLP with smooth activation and cross-entropy loss. Unlike ReLU activation or DLR loss, which might create degeneration edges, the chain rule is readily applied in this case, and the Lipschitz constant of the loss is naturally computed by estimating its gradients. Similar to the analysis of the ReLU networks, we obtain the upper bound of the WDRO as $L \triangleq 2^{1/s} \max_{x \in \mathcal{X}} \|\nabla \theta(x)\|_{r \to s}$ while the worst-case distribution and lower bound are constructed similar to the ReLU networks.

3. Finally, we bridge the gap between WDRO theory and adversarial evaluation by introducing the Wasserstein Distributional Attack (WDA), which directly constructs adversarial distributions within the Wasserstein ball rather than restricting to point-wise perturbations. Unlike existing attacks that place all adversarial examples on the $\epsilon$-ball boundary, WDA flexibly interpolates between point-wise ($\kappa = 1$) and truly distributional attacks ($\kappa > 1$) by supporting adversarial distributions on $2N$ points. This offers a complementary perspective to strong baselines such as AutoAttack and the RobustBench leaderboard (Croce & Hein, 2020; Croce et al., 2021). Empirically, WDA with $\kappa = 2$ consistently finds stronger adversarial examples than state-of-the-art methods across diverse settings: achieving lower robust accuracy than APGD-DLR on CIFAR-10/100 with WideResNet backbone (Zagoruyko & Komodakis, 2016) on both $\ell_\infty$ and $\ell_2$ perturbations. When integrated into the Adaptive Auto Attack framework, our method matches or exceeds the ensemble performance of A[3]. These results demonstrate that the distributional perspective not only provides tighter theoretical certificates but also yields more effective attacks, validating our claim that existing robustness evaluations underestimate vulnerability by restricting to $\Omega_\infty$ rather than the larger $\Omega_1$ ambiguity set assumed by certificates.

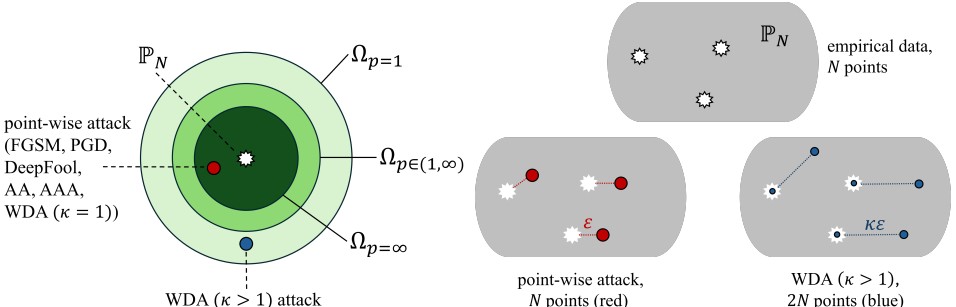

Figure 1: **Left**: Wasserstein ambiguity ball $\Omega_p = \{\mathbb{P} : \mathcal{W}_{d,p}(\mathbb{P}, \mathbb{P}_N) \le \epsilon\}$ inclusion and its admissible attacks. Our proposed Wasserstein Distributional Attack (WDA) with $\kappa \ge 1$ includes its special case $\kappa = 1$ as a point-wise attack, and produces a distributional attack when $\kappa > 1$. Note that most of the existing tight certificates estimated an upper bound of WDRO w.r.t. $\Omega_{p=1}$, not $\Omega_{p=\infty}$. **Right**: Visualization of point-wise attack ($N$ adversarial samples) versus our WDA ($2N$ adversarial samples). Our WDA allows not only a larger number of supports but also a wider range of perturbations.

## 2 PRELIMINARIES

**Notations** We denote basis vector as $\boldsymbol{e}_k$; indicator function as $\mathbf{1}_{\{\cdot\}}$; Dirac measure as $\boldsymbol{\delta}_z$; input dimension $n$, and output dimension as $K$. An empirical dataset is denoted $\{Z^{(1)}, \ldots, Z^{(N)}\}$ with $Z = (x, y) \in \mathcal{Z} = \mathcal{X} \times \mathcal{Y}$ where $\mathcal{X} \subset \mathbb{R}^n$ and $\mathcal{Y} \subset \mathbb{R}^K$; empirical distribution $\mathbb{P}_N = \sum_i \mu_i \boldsymbol{\delta}_{Z^{(i)}}$ with $Z^{(i)} = (x^{(i)}, y^{(i)} = \boldsymbol{e}_{k_i})$. Norms $\|\cdot\|_r$ and $\|\cdot\|_s$ are dual with $1/r + 1/s = 1$. For a matrix $A$, $\|A\|_{r \to s} = \sup_{\|u\|_r = 1} \|Au\|_s$. Rectifier $[\cdot]_+ = \max\{0, \cdot\}$. Recession cone $\text{rec}(\cdot)$.

Interior set $\text{int}(\cdot)$. Ground cost $d((x', y'), (x, y)) = \|x' - x\|_r + \infty \cdot \mathbf{1}_{\{y' \neq y\}}$. Cross-entropy loss $\ell(x, y; \theta) = -\sum_{k=1}^{K} y_k \log \text{softmax}(\theta(x))_k$. Analogous DLR loss as defined in Croce & Hein (2020). We define the dual-norm maximizer $\mathcal{M}_r$ by

$$\mathcal{M}_r \colon g \mapsto \arg\max_h \{\langle g, h \rangle \mid \|h\|_r = 1\} = \begin{cases} \text{sign}(g) & \text{if } r = \infty, \\ g/\|g\|_2 & \text{if } r = 2, \\ \text{sign}(g_{k'})\boldsymbol{e}_{k'} \text{ with } k' \in \arg\max_k |g_k| & \text{if } r = 1, \end{cases} \tag{1}$$

and projection $\Pi_{r,x,\kappa\epsilon}$ by

$$\Pi_{r,x,R} \colon x \mapsto \arg\min_\xi \left\{ \|\xi - x\|_2^2 \mid \|\xi\|_r \leq R \right\}. \tag{2}$$

**Wasserstein Distributionally Robust Optimization (WDRO)**  Robustness guarantees and certificates aim to make model predictions trustworthy under adversarial manipulation (Wong & Kolter, 2018; Cohen et al., 2019; Salman et al., 2019). The empirical risk minimization model $\inf_\theta \mathbb{E}_{\mathbb{P}_N}[\ell(Z; \theta)]$ optimizes average performance on the observed data but offers no protection against worst-case shifts nearby. Distributionally robust optimization (DRO) addresses this by choosing parameters that perform well against all distributions within a prescribed neighborhood: $\inf_\theta \sup_{\mathbb{P} \in \mathcal{P}} \mathbb{E}_{\mathbb{P}}[\ell(Z; \theta)]$. Here, the worst-case loss is taken over all admissible distributions $\mathbb{P} \in \mathcal{P}$. The ambiguity (or uncertainty) set $\mathcal{P}$ is often constructed by collecting all distributions $\mathbb{P}$ that are similar to the empirical distribution $\mathbb{P}_N$.

In this work, we focus on the Wasserstein ambiguity set, which is a ball centered at $\mathbb{P}_N$ under the Wasserstein distance. Given a ground cost $d$ on the space of data $\mathcal{Z}$, the Wasserstein distance (Villani et al., 2008) between two distributions $\mathbb{P}$ and $\mathbb{Q}$ it is defined as $\mathcal{W}_{d,p}(\mathbb{P}, \mathbb{Q}) \triangleq \left( \inf_{\pi \in \Pi(\mathbb{P}, \mathbb{Q})} \int_{\mathcal{Z} \times \mathcal{Z}} d^p(z', z) \, \mathrm{d}\pi(z', z) \right)^{1/p}$ for $p \in [1, \infty)$;, and $\mathcal{W}_{d,p}(\mathbb{P}, \mathbb{Q}) \triangleq \inf_{\pi \in \Pi(\mathbb{P}, \mathbb{Q})} \text{ess.} \sup_\pi(d)$ for $p = \infty$. Intuitively, the Wasserstein distance between two distributions $\mathbb{P}$ and $\mathbb{Q}$ is defined as the minimum cost to transport the mass of $\mathbb{P}$ to $\mathbb{Q}$. The WDRO problem with a given budget of perturbation $\epsilon > 0$ can be written as

$$\inf_\theta \sup_{\mathbb{P} \in \Omega_p} \mathbb{E}_{\mathbb{P}}[\ell(Z; \theta)] \text{ where } \Omega_p = \{\mathbb{P} \mid \mathcal{W}_{d,p}(\mathbb{P}, \mathbb{P}_N) \leq \epsilon\}. \tag{3}$$

It is worth noting that $\mathcal{W}_{d,p} \leq \mathcal{W}_{d,p'}$ if $p \leq p'$, thus $\Omega_1 \supseteq \Omega_p \supseteq \Omega_{p'} \supseteq \Omega_\infty$ (see Figure 1). For more details of Wasserstein distributionally robust optimization, we refer reader to Kuhn et al. (2019) and our Appendix A.

**Lipschitz Certificate**  For $p = 1$, the worst-case risk over a Wasserstein ball admits the standard Lipschitz upper bound

$$\sup_{\mathbb{P} \in \Omega_1} \mathbb{E}_{\mathbb{P}}[\ell(Z; \theta)] \leq \mathbb{E}_{\mathbb{P}_N}[\ell(Z; \theta)] + L\epsilon. \tag{4}$$

where $L$ is any Lipschitz constant of $z \mapsto \ell(z; \theta)$ with respect to the ground cost. This inequality follows from weak duality and is widely used to make the WDRO objective tractable: one replaces the inner maximization by the surrogate $L\epsilon$ and then controls $L$ (Mohajerin Esfahani & Kuhn, 2018; Blanchet et al., 2019; Gao & Kleywegt, 2023; Gao et al., 2024). In practice, estimating $L$ reduces to bounding the network's (global or local) Lipschitz modulus, e.g., fast global products of per-layer operator norms (Virmaux & Scaman, 2018) or tighter activation-aware/local certificates (Jordan & Dimakis, 2020; Shi et al., 2022).

**Adversarial Attack.**  Adversarial attack methods often construct a perturbed distribution by shifting each sample $X^{(i)}$ along a specific adversarial direction $u^{(i)}$ to get $X_{\text{adv}}^{(i)}$ (Goodfellow et al., 2014; Moosavi-Dezfooli et al., 2016; Carlini & Wagner, 2017). These methods are essentially point-wise attacks, which draws a distribution $\mathbb{P}_{\text{adv}} = \sum_{i=1}^{N} \frac{1}{N} \boldsymbol{\delta}_{X_{\text{adv}}^{(i)}}$ in the Wasserstein ambiguity set $\Omega_p = \{\mathbb{P} \colon \mathcal{W}_{d,p} \leq \epsilon\}$ when $p = \infty$ (see Figure 1). Whereas, in the $p = 1$ case, the ambiguity set only constrains the average transportation cost under an optimal coupling. Hence, the adversary may move some points farther and others less as long as the mean cost stays within budget. This creates a significant gap between the robustness measured against $\Omega_{p=\infty}$ attacks and the theoretical robustness or Lipschitz certificates 4 which are developed for $\Omega_{p=1}$ (Mohajerin Esfahani & Kuhn, 2018; Carlini et al., 2019; Rice et al., 2021).

## 3 TRACTABLE INTERPRETATION OF WDRO FOR NEURAL NETWORKS

For certain shallow and convex models (e.g., linear regression, support vector machines, etc.), the tractable representation of the WDRO problem 3 is well-established in the literature (Mohajerin Esfahani & Kuhn, 2018; Blanchet et al., 2019; Gao & Kleywegt, 2023; Gao et al., 2024). This tractable form enables a computational advantage and provides a clear interpretation of the robustness of regularization mechanisms. In that line of work, the Lipschitz constant often provides a practical and tight upper bound of the corresponding upper bounds. However, when the loss is non-convex, the Lipschitz certificate is not always tight, as outlined in the following remark.

*Remark.* Consider a single-point empirical $\mathbb{P}_N = \boldsymbol{\delta}_{\{X^{(1)}=2\}}$ and a loss given by

$$
\ell(x) = \begin{cases} |x| & \text{if } |x| \leq 1, \\ \frac{1}{2}|x| + \frac{1}{2} & \text{otherwise.} \end{cases}
$$

Then $\ell$ is Lipschitz with modulus 1, however $\sup_{\Omega_1} \mathbb{E}_{\mathbb{P}}[\ell(X)] = \mathbb{E}_{\mathbb{P}_N}[\ell(X)] + \frac{\epsilon}{2}$ for any $\epsilon > 0$.

As presented in the following sections, our main theoretical results (Theorem 3.1 and 3.3) show that Lipschitz modulus provides a tight upper bound for the WDRO problem 3 for a class of ReLU neural networks and smooth activated neural networks.

### 3.1 EXACT AND TRACTABLE INTERPRETATION OF WDRO FOR RELU NEURAL NETWORKS

For a broad class of ReLU networks, the tight (local) Lipschitz constant can be found exactly via activation patterns. For example, for any $H$-layer ReLU network $\theta(x) = W_{H+1}(\text{ReLU}(\cdots(W_1 x + b_1)\cdots) + b_H)$, let

$$
L_\theta = \sup_{x \in \mathcal{X}} \sup_{J \in \partial\theta(x)} \|J\|_{r \to \tilde{r}}, \tag{5}
$$

where $J \in \partial\theta(x)$ is a general Jacobian of $\theta$ at $x$, then Jordan & Dimakis (2020, Theorem 1) has shown that $\|\theta(x') - \theta(x)\|_{\tilde{r}} \leq L_\theta \times \|x' - x\|_r$ for any $x', x \in \mathcal{X}$. Moreover, if $\theta$ is in general position (Jordan & Dimakis, 2020, Definition 4), then the chain rule applies and any general Jacobian $J$ must has a form as $W_{H+1}D_H W_H \cdots D_1 W_1$ for some $[0,1]$-diagonal matrix $D_h$, $h = 1, \ldots, H$. It is worth noting that the set of ReLU networks *not* in general position is negligible (Jordan & Dimakis, 2020, Theorem 3). Now in equation 5, the maximizer of a convex function (norm operator) must happen at vertices, thus we only need to consider $0/1$-diagonal matrix $D_h$.

We formally introduce the concept of mask as follows.

**Definition 3.1** (Mask and Cell). Let $\theta(x) = W_{H+1}(\text{ReLU}(\cdots(W_1 x + b_1)\cdots) + b_H)$ be a ReLU network which is in general position. For any tuple $\boldsymbol{D} = (D_1, \ldots, D_H)$, we define

$$
J_{\boldsymbol{D}} = W_{H+1}D_H W_H \cdots D_1 W_1.
$$

For any $x \in \mathcal{X}$, we define the set of all $0/1$-diagonal masks at $x$ as

$$
\mathcal{D}_x = \{\boldsymbol{D} = (D_1, \ldots, D_H) \mid J_{\boldsymbol{D}} \in \partial\theta(x),\ D_h \text{ is } 0/1\text{-diagonal},\ h = 1, \ldots, H\}
$$

We denote $\mathcal{D}_{\mathcal{X}} = \cup_{x \in \mathcal{X}} \mathcal{D}_x$ as the (finite) set of all possible masks.

For any mask $\boldsymbol{D} = (D_1, \ldots, D_H) \in \mathcal{D}_x$, let $\mathcal{C}_{\boldsymbol{D}}$ be the cell, which is an open linear region, defined by

$$
\mathcal{C}_{\boldsymbol{D}} = \{x \mid \text{pre}_h(x)_j > 0 \text{ if } D_h(j,j) = 1 \text{ and } \text{pre}_h(x)_j < 0 \text{ if } D_h(j,j) = 0,\ h = 1, \ldots, H\},
$$

where the pre-activation functions are defined as

$$
\text{pre}_h \colon x \mapsto W_h(\text{ReLU}(\cdots(W_1 x + b_1)\cdots) + b_h).
$$

Given this definition of mask and note that $\mathcal{D}_{\mathcal{X}}$ is finite, one can rewrite equation 5 as $L_\theta = \max_{\boldsymbol{D} \in \mathcal{D}_{\mathcal{X}}} \|J_{\boldsymbol{D}}\|_{r \to \tilde{r}}$. We adopt this notion and show that it induces an upper bound for the Wasserstein distributional robust optimization (WDRO) problem 3 with cross-entropy loss. Moreover, this upper bound is tight for a class of monotonic ReLU networks.

**Theorem 3.1** (WDRO for ReLU). *Given a ReLU network $\theta(x) = W_{H+1}(\text{ReLU}(\cdots(W_1 x + b_1)\cdots) + b_H)$ being in general position, $1/r + 1/s = 1$ and $\ell$ being the cross-entropy or DLR loss, define*

$$\boldsymbol{L} \triangleq 2^{1/s} \max_{\boldsymbol{D} \in \mathcal{D}_{\mathcal{X}}} \|J_{\boldsymbol{D}}\|_{r \to s}, \tag{6}$$

*and*

$$\boldsymbol{l} \triangleq \max_{\substack{x \in \mathcal{X}, \ k' \neq k \\ \boldsymbol{D} \in \mathcal{D}_x}} \sup_{\|u\|_r = 1} \left\{ (\boldsymbol{e}_{k'} - \boldsymbol{e}_k)^\top J_{\boldsymbol{D}} u \mid u \in \text{rec}(\mathcal{C}_{\boldsymbol{D}}) \right\}. \tag{7}$$

*where $J_{\boldsymbol{D}}, \mathcal{C}_{\boldsymbol{D}}, \mathcal{D}_{\mathcal{X}}$ are defined in Definition 3.1 and $\text{rec}(\mathcal{C}_{\boldsymbol{D}})$ is the recession cone of $\mathcal{C}_{\boldsymbol{D}}$. Then for any $\epsilon > 0$, we have*

$$\mathbb{E}_{\mathbb{P}_N}[\ell(Z; \theta)] + \boldsymbol{l}\epsilon \leq \sup_{\mathbb{P}: \ \mathcal{W}_{d,1}(\mathbb{P}, \mathbb{P}_N) \leq \epsilon} \mathbb{E}_{\mathbb{P}}[\ell(Z; \theta)] \leq \mathbb{E}_{\mathbb{P}_N}[\ell(Z; \theta)] + \boldsymbol{L}\epsilon. \tag{8}$$

*Moreover, if the dual-norm maximizer $\mathcal{M}_r(J_{\boldsymbol{D}^\star}^\top (\boldsymbol{e}_{k'^\star} - \boldsymbol{e}_{k^\star})) \in \text{rec}(\mathcal{C}_{\boldsymbol{D}^\star})$ where $\boldsymbol{D}^\star$ is a maximizer of 6 and $(k'^\star, k^\star)$ is a maximizer of 7, and $(\boldsymbol{e}_{k'^\star} - \boldsymbol{e}_{k^\star})$ is the largest increment direction of $J_{\boldsymbol{D}^\star}$, then $\boldsymbol{l} = \boldsymbol{L}$.*

*Proof.* To prove inequality 8, we show that $\ell(\cdot, \theta)$ is $\boldsymbol{L}$-Lipschitz, and a direction $u$ found in equation 7 induces an admissible attack $\mathbb{P}_{\text{adv}}$ satisfying that $\mathbb{E}_{\mathbb{P}_{\text{adv}}}[\ell(Z; \theta)] \approx \mathbb{E}_{\mathbb{P}_N}[\ell(Z; \theta)] + \boldsymbol{l}\epsilon$ and $\mathcal{W}_{d,1}(\mathbb{P}_{\text{adv}}, \mathbb{P}_N) \leq \epsilon$. To verify the sufficient condition of $\boldsymbol{l} = \boldsymbol{L}$, we show that the constructed $\mathbb{P}_{\text{adv}}$ provides $\boldsymbol{l} = \boldsymbol{L}$. We provide detailed proof in Appendix B.1. $\qquad\square$

In Figure 2a, we illustrate an instance in which our lower and upper bounds match. While equation 7 provides a tight lower bound of the WDRO, it is impractical to scan through all $x \in \mathcal{X}$ and its mask $\mathcal{D}_x$. We then introduce a practical lower bound, of which we consider the mask associated with the sample points only.

**Corollary 3.2** (Practical lower bound). *Given assumptions and notations used in Theorem 3.1, let $\mathcal{Z}_N = \{(X^{(1)}, Y^{(1)}), \dots, (X^{(N)}, Y^{(N)})\}$ and*

$$\boldsymbol{l}_N \triangleq \max_{\substack{(X^{(i)}, Y^{(i)}) \in \mathcal{Z}_N, \\ \boldsymbol{D} \in \mathcal{D}_x}} \max_k \sup_{\|u\|_r = 1} \left\{ (\boldsymbol{e}_k - Y^{(i)})^\top J_{\boldsymbol{D}} u \mid u \in \text{int}(\text{rec}(\mathcal{C}_{\boldsymbol{D}})) \right\}. \tag{9}$$

*Then $\boldsymbol{l}_N \leq \boldsymbol{l}$.*

Based on the proof of our lower bound (equation 16), we construct a worst-case distribution by moving mass from a sample along a direction $u$ that maximizes the margin term in equation 7. In Section 4, based on formulation 9, we create this construction empirically via the attack distribution equation 10 by choosing adversarial direction $u^{(i)}$ for each sample $i$ so that it maximizes the first-order increase of the corresponding logit margin.

## 3.2 EXACT AND TRACTABLE INTERPRETATION OF WDRO FOR SMOOTH ACTIVATION NEURAL NETWORKS

For networks with smooth activations, e.g, GELU (Hendrycks & Gimpel, 2016), SiLU/Swish (Ramachandran et al., 2017; Elfwing et al., 2018), WDRO duality connects worst-case (adversarial) risk to first-order geometry via the Jacobian of the logit map, yielding global Lipschitz-type upper penalties of the form $\sup_{x \in \mathcal{X}} \|J(x)\|_{r \to s}$. Compared to piecewise-linear ReLU certificates, smooth nets trade exact cell-wise constancy for differentiability along rays and curves, suggesting bounds driven by asymptotic Jacobian behavior rather than activation masks.

Let $t$ be a positive scalar, $\theta : \mathcal{X} \subseteq \mathbb{R}^n \to \mathbb{R}^K$ be a classifier with smooth activations and cross-entropy loss; let $J(x) \in \mathbb{R}^{K \times n}$ be its Jacobian. We then have the following result.

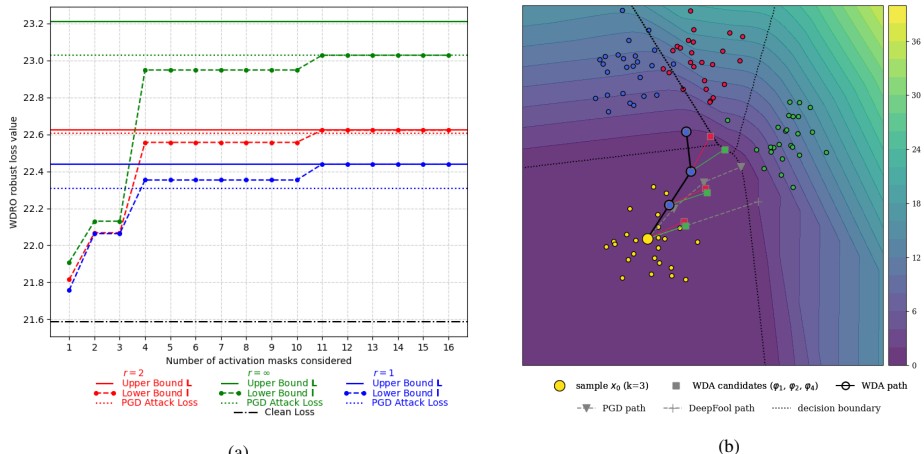

(a)    (b)

Figure 2: (a) WDRO bounds and PGD attack loss for a fixed $n = K = 2$ ReLU classifier with one hidden layer of dimension 8. Lower-bound curves are the cumulative $l$ as more reachable activation masks are considered. (b) Wasserstein Distributional Attack (WDA, Alg. 1) for $r = 2$. At each iteration $x_t$, WDA forms $K-1$ candidates $\varphi_j$ and updates using the one with the largest logit $\theta_j(\varphi_j)$. For reference, PGD follows the dual-norm gradient direction; DeepFool linearizes the decision boundary.

**Theorem 3.3** (WDRO for Smooth Networks). *Let $\theta : \mathbb{R}^n \to \mathbb{R}^k$ be a differentiable network, $1/r + 1/s = 1$ and $\ell$ being the cross-entropy or DLR loss, define*

$$L \triangleq 2^{1/s} \sup_{x \in \mathcal{X}} \|\nabla_x \theta(x)\|_{r \to s},$$

*and*

$$l \triangleq \sup_{x \in \mathcal{X}} \max_{k' \neq k} \sup_{\|u\|_r = 1} \left\{ (e_{k'} - e_k)^\top \nabla_x \theta(x) u \right\}.$$

*Then for any $\epsilon > 0$,*

$$\mathbb{E}_{\mathbb{P}_N}[\ell(Z;\theta)] + l\epsilon \leq \sup_{\mathbb{P} : \mathcal{W}_{d,1}(\mathbb{P}, \mathbb{P}_N) \leq \epsilon} \mathbb{E}_{\mathbb{P}}[\ell(Z;\theta)] \leq \mathbb{E}_{\mathbb{P}_N}[\ell(Z;\theta)] + L\epsilon.$$

In this setting, first-order WDRO penalties are controlled by how $J(x)$ amplifies unit directions and how that amplification projects onto the most competitive non-true logit. The upper slope $L$ is the global worst-case amplification, while the lower, margin–directional slope $l$ follows rays $x^{(i)} + tu$ and harvests only the component along $(e_{k'} - e_{k_i})$. When a ray both attains the global operator norm and aligns with a margin difference, the bound is tight to first order ($l = L$). This motivates the adversarial procedure used in our adversarial attack algorithm WDA (Algorithm 1), where we search for a direction $u$ and a rival class $k'$ that maximize the first-order increase $(e_{k'} - e_{k_i})^\top J(x)u$.

## 4 WASSERSTEIN DISTRIBUTIONAL ATTACK

Existing point-wise attacks such as FGSM (Goodfellow et al., 2014), DeepFool (Moosavi-Dezfooli et al., 2016), AA (Croce & Hein, 2020), AAA (Liu et al., 2022), keep the adversarial distribution supported on exactly $N$ points, where each point $X_{\text{adv}}^{(i)}$ is perturbed to be precisely on the boundary of the $\epsilon$-ball centered at $X^{(i)}$. To address this issue, we propose a novel method called *Wasserstein Distributional Attack (WDA)*. At a high level, WDA constructs an adversarial distribution, $\mathbb{P}_{\text{adv}}$, supported on a set of $2N$ points. This set consists of $N$ original empirical samples $X^{(i)}$ and $N$ corresponding adversarial points $X_{\text{adv}}^{(i)}$, each perturbed to an $r$-norm distance $\kappa\epsilon$ from $X^{(i)}$ using the first-order, margin-aligned directions predicted by Theorems 3.1–3.3, for some $\kappa \geq 1$. In other words,

$$\mathbb{P}_{\text{adv}} = \frac{1}{N} \sum_{i=1}^{N} \left( \left(1 - \frac{1}{\kappa}\right) \boldsymbol{\delta}_{(X^{(i)}, Y^{(i)})} + \frac{1}{\kappa} \boldsymbol{\delta}_{(X_{\text{adv}}^{(i)}, Y^{(i)})} \right). \tag{10}$$

In the special case where $\kappa = 1$, our proposed attack reduces to existing point-wise methods. When $\kappa = 2$, WDA simplifies to a uniform distribution over all $2N$ points, with each point receiving a weight of $\frac{1}{2N}$. This $2N$-support mixture belongs to the $\Omega_1$ ambiguity set and serves as a constructive, distributional adversary; it is not necessarily the inner maximizer of WDRO. We now make the first-order ascent directions explicit; this is the step used by WDA to realize the margin-aligned rays from Theorems 3.1–3.3.

Define (sub)gradient $g_j(x) \in \partial_x(\theta_j - \theta_k)(x)$. Then $\mathcal{M}_r(g_j)$ give the per–iteration, first–order version of the ray ascent used in Theorems 3.1–3.3: within a ReLU cell (affine logits) or for smooth activations (continuous $J$), moving along $u_j = \mathcal{M}_r(g_j)$ increases the gap at rate $\|g_j(x)\|_s$. During an initial probing phase, we evaluate all rivals $j \neq k$ using these first-order steps. At the end of that phase, we fix a single rival $j^*$ based on the logits magnitude and continue the remaining iterations. If we allow $j^*$ to change at every step, the update can oscillate across classes and chase locally steep but globally suboptimal directions for misclassifications. Finally, we project each step to the ball of radius $\kappa\varepsilon$ around the anchor $X^{(i)}$ to the WDRO budget. The procedure for implementing the Wasserstein Distributional Attack is presented in Algorithm 1. A visualization of our algorithm is shown in Figure 2b.

---

**Algorithm 1** Wasserstein Distributional Attack (WDA)

---

1: **Inputs:** neural network $\theta : \mathbb{R}^n \to \mathbb{R}^K$, empirical distribution $\mathbb{P}_N = \sum_{i=1}^N \boldsymbol{\delta}_{(X^{(i)}, Y^{(i)})}$, budget $\epsilon > 0$, cost-norm
    $r \in \{1, 2, \infty\}$, WDA parameter $\kappa \geq 1$, step size $\alpha > 0$, and $0 < \text{prob} \leq \text{maxiter}$
2: **Outputs:** Wasserstein distributional attack $\mathbb{P}_{\text{adv}}$ such that $\mathcal{W}_{d,1}(\mathbb{P}_{\text{adv}}, \mathbb{P}_N) \leq \epsilon$ where $d((x', y'), (x, y)) = \|x' - x\|_r + \infty \cdot \mathbf{1}_{\{y' \neq y\}}$
3: **Initialize:** dual-norm maximizer $\mathcal{M}$ (1), projection $\Pi$ (2)
4: **for** $i = 1$ to $N$ **do**
5:     $x_0 \leftarrow X^{(i)}$, $\boldsymbol{e}_k \leftarrow Y^{(i)}$ for some $k = 1, \ldots, K$
6:     **for** iter $= 0$ to maxiter **do**
7:         **if** iter $<$ prob **then** $\mathcal{J} = \{1, \ldots, K\} \setminus \{k\}$ **else** $\mathcal{J} = \{j^*\}$
8:         $g_j \leftarrow \nabla_x \theta(x_{\text{iter}})^\top (\boldsymbol{e}_j - \boldsymbol{e}_k)$ for $j \in \mathcal{J}$
9:         $u_j \leftarrow \mathcal{M}_r(g_j)$ for $j \in \mathcal{J}$
10:        $\varphi_j \leftarrow \Pi_{r, X^{(i)}, \kappa\epsilon}(x_{\text{iter}} + \alpha u_j)$ for $j \in \mathcal{J}$
11:        $j^* = \arg\max_{j \in \mathcal{J}} \theta_j(\varphi_j)$
12:        $x_{\text{iter}+1} \leftarrow \varphi_{j^*}$
13:     **end for**
14:     $X_{\text{adv}}^{(i)} \leftarrow x_{\text{maxiter}}$
15: **end for**
16: $\mathbb{P}_{\text{adv}} \leftarrow \frac{1}{N} \sum_{i=1}^N \left(1 - \frac{1}{\kappa}\right) \boldsymbol{\delta}_{(X^{(i)}, Y^{(i)})} + \frac{1}{\kappa} \boldsymbol{\delta}_{(X_{\text{adv}}^{(i)}, Y^{(i)})}$
17: **return** $\mathbb{P}_{\text{adv}}$

---

## 5   Related Work

**Robustness Certificates**    Early scalable global certificates control the Lipschitz constant by multiplying per-layer operator norms, which is fast to compute yet data-agnostic and typically loose on deep nets (Virmaux & Scaman, 2018). For ReLU networks, local (activation-aware) methods exploit piecewise linearity to produce much tighter, input-conditioned certificates on individual activation regions (Katz et al., 2017; Ehlers, 2017; Weng et al., 2018; Singh et al., 2018; Shi et al., 2022). Most relevant to exact local Lipschitzness, Jordan & Dimakis (2020) showed that for a broad class of ReLU networks in general position, the local Lipschitz constant can be computed exactly by optimizing over activation patterns.

**Adversarial Attacks**    Adversarial Attack methods seek for perturbation $x'$ formed by adding a small, human-imperceptible perturbation to a clean input $x$ that causes misclassification (Szegedy et al., 2014). The threat model specifies the attacker's knowledge (white-box vs. black-box), the admissible perturbation set (e.g., $r_2$ balls with budget $\epsilon$), and the objective (e.g., worst-case loss within the ball). Canonical white-box methods include FGSM (Goodfellow et al., 2014), multistep PGD (Madry et al., 2018), CW (Carlini & Wagner, 2017), and gradient-based margin attacks such as DeepFool (Moosavi-Dezfooli et al., 2016). Decision-based and score-free attacks (black-box) include Boundary Attack (Brendel et al., 2021) and Square Attack (Andriushchenko et al., 2020). Robust evaluation is subtle: gradient masking can inflate apparent robustness if attacks are not adapted (Athalye et al., 2018).

To standardize evaluation, AutoAttack (AA) (Croce & Hein, 2020) composes strong, parameter-free attacks (APGD-CE, APGD-DLR, FAB, Square) and is widely adopted for reporting robust accuracy. RobustBench (Croce et al., 2021) curates model zoos and standardized test protocols across datasets and $r_p$ threat models, enabling comparable and reproducible robustness claims. Liu et al. (2022) proposed Adaptive Auto Attack ($A^3$), which incorporates Adaptive Direction Initialization (ADI) and Online Statistics-based Discarding (ODS) (Tashiro et al., 2020) to enhance attack efficiency. In our experiments, we report robustness under AA and $A^3$ following RobustBench conventions and use them as baselines for comparison.

Several works have focused on adversarial attacks tailored to ReLU networks. Croce & Hein (2018) introduced rLR-QP, a gradient-free method that navigates the piecewise-linear regions of ReLU models by solving convex subproblems and enhancing exploration with randomization and local search. More recently, Zhang et al. (2022) developed BaB-Attack, a branch-and-bound framework that operates in activation space, leveraging bound propagation, beam search, and large neighborhood search to uncover stronger adversarial examples than conventional gradient-based approaches, particularly on hard-to-attack inputs. As pointed out in Zhang et al. (2022); Croce et al. (2020), rLR-QP and BaB-Attack are not as efficient as gradient based attack, therefore, we only use APGD as single-method baseline in our experiment.

Table 1: Comparison of robust accuracy of WDA and baseline methods against various defenses on CIFAR-10, CIFAR-100 and ImageNet. The best (lowest) attack accuracy of single methods and ensemble methods are highlighted in underline and **bold**, respectively.

| PAPER | MODEL | CLEAN | SINGLE METHOD | | | | ENSEMBLE METHOD | | |
|---|---|---|---|---|---|---|---|---|---|
| | | | APGD-CE | APGD-DLR | WDA ($\kappa=1$) | WDA ($\kappa=2$) | AA | $A^3$ | $A^3$++ |
| CIFAR-10 – $r_\infty$, $\epsilon = 8/255$ | | | | | | | | | |
| BARTOLDSON ET AL. (2024) | WRN-94-16 | 93.68 | 76.15 | 74.31 | 74.05 | 65.25 | 73.71 | 73.55 | **73.54** |
| BARTOLDSON ET AL. (2024) | WRN-82-8 | 93.11 | 74.17 | 72.54 | 71.85 | 62.06 | 71.59 | 71.46 | **71.46** |
| CUI ET AL. (2024) | WRN-28-10 | 92.16 | 70.60 | 68.62 | 68.07 | 60.01 | 67.73 | 67.58 | **67.57** |
| WANG ET AL. (2023) | WRN-70-16 | 93.25 | 73.46 | 71.68 | 71.02 | 63.08 | 70.69 | 70.53 | **70.52** |
| WANG ET AL. (2023) | WRN-28-10 | 92.44 | 70.24 | 68.24 | 67.60 | 60.96 | 67.31 | 67.17 | **67.17** |
| XU ET AL. (2023) | WRN-28-10 | 93.69 | 67.08 | 69.00 | 66.39 | 63.25 | 63.89 | 63.93 | **63.84** |
| SEHWAG ET AL. (2022) | RN-18 | 84.59 | 58.40 | 57.66 | 56.30 | 54.65 | 55.54 | 55.50 | **55.50** |
| CIFAR-10 – $r_2$, $\epsilon = 0.5$ | | | | | | | | | |
| WANG ET AL. (2023) | WRN-70-16 | 95.54 | 85.66 | 85.30 | 85.00 | 77.63 | 84.97 | **84.96** | 84.97 |
| WANG ET AL. (2023) | WRN-28-10 | 95.16 | 84.52 | 83.89 | 83.71 | 76.31 | 83.68 | 83.68 | **83.68** |
| SEHWAG ET AL. (2022) | WRN-34-10 | 90.93 | 78.23 | 78.16 | 77.51 | 72.01 | 77.24 | **77.22** | 77.25 |
| SEHWAG ET AL. (2022) | RN-18 | 89.76 | 75.24 | 75.32 | 74.69 | 69.75 | 74.41 | 74.41 | **74.40** |
| DING ET AL. (2020) | WRN-28-4 | 88.02 | 66.62 | 66.62 | 66.22 | 63.04 | 66.09 | **66.05** | 66.06 |
| CUI ET AL. (2024) | WRN-28-10 | 89.05 | 66.58 | 67.08 | 66.59 | 64.19 | 66.44 | **66.41** | 66.42 |
| CIFAR-100 – $r_\infty$, $\epsilon = 8/255$ | | | | | | | | | |
| WANG ET AL. (2023) | WRN-28-10 | 72.58 | 44.09 | 39.66 | 39.12 | 43.61 | 38.77 | **38.70** | 38.71 |
| ADDEPALLI ET AL. (2022) | RN-18 | 65.45 | 33.47 | 28.82 | 28.26 | 37.64 | 27.67 | 27.65 | **27.63** |
| CUI ET AL. (2024) | WRN-28-10 | 73.85 | 43.82 | 40.37 | 39.57 | 43.68 | 39.18 | 39.17 | **39.14** |
| IMAGENET – $r_\infty$, $\epsilon = 4/255$ | | | | | | | | | |
| LIU ET AL. (2025) | CONVNEXT-B | 76.02 | 55.90 | 56.78 | 54.38 | 52.95 | 55.82 | 53.19 | **53.18** |
| SINGH ET AL. (2023) | CVNEXT-S-CVST | 74.10 | 52.82 | 53.20 | 51.04 | 50.31 | 52.42 | 49.92 | **49.90** |

## 6 EXPERIMENTS

**Experimental settings** To evaluate the effectiveness of WDA, we test the adversarial robustness of several state-of-the-art defense models on CIFAR-10, CIFAR-100 and ImageNet. We report $r_\infty$ and $r_2$ robustness under perturbation budgets $\epsilon \in \{4/255, 8/255, 0.5\}$. In addition to point-wise attack, we conduct a separate Wasserstein distributional attack experiment to further validate our method. Specifically, we set $\kappa = 2$ in Algorithm 1 to find the adversarial (distributional) attack $\mathbb{P}_{adv}$ (equation 10) and reporting classification accuracy on the distribution by $(1 - 1/\kappa) \times \text{acc}_{clean} + (1/\kappa) \times \text{acc}_{adv}$. Our attack is benchmarked against AA, APGD-DLR, APGD-CE (Croce & Hein, 2020), and $A^3$ (Liu et al., 2022). The defense models, along with their official implementations and pretrained weights, are obtained from RobustBench Croce et al. (2021). All experiments are conducted on 2x NVIDIA GeForce RTX 4090 GPU and 1x NVIDIA H200.

### 6.1 COMPARISON WITH EXISTING BASELINES

**Setup** For the baselines AA, APGD-DLR, APGD-CE, and $A^3$, we adopt the configurations reported in their respective research papers. Meanwhile, in WDA, we set the number of probe steps to $\alpha_{probe} = 10$, and use $\alpha_{atk} = 20$ attack iterations. We further propose $A^3$++, an extension of $A^3$ that incorporates our attack into its framework.

**Robustness on $r_\infty$ and $r_2$ Perturbations for CIFAR-10** Table 1 presents the robust accuracy of several attack methods under $r_\infty$ perturbations with $\epsilon = 8/255$ and $r_2$ perturbations with $\epsilon = 0.5$. Across both single and ensemble based threat models, WDA consistently outperforms other single-method attacks (APGD-CE and APGD-DLR), highlighting its effectiveness as a stronger standalone adversarial evaluation. Moreover, WDA achieves results that are often comparable to ensemble-based methods, indicating its ability to match the strength of more computationally demanding attack aggregations. Within the ensemble family, $A^3$++ demonstrates clear improvements over AA and provides competitive performance with $A^3$, surpassing it on several defense models (3 out of 7 under $r_\infty$ and 1 out of 6 under $r_2$). Notably, under $r_\infty$ and $r_2$, WDA ($\kappa = 2$) produces lower robust accuracy values than other attacking methods across all defense models. This highlights the potential of Wasserstein distributionally attack.

### 6.2 ABLATION STUDY

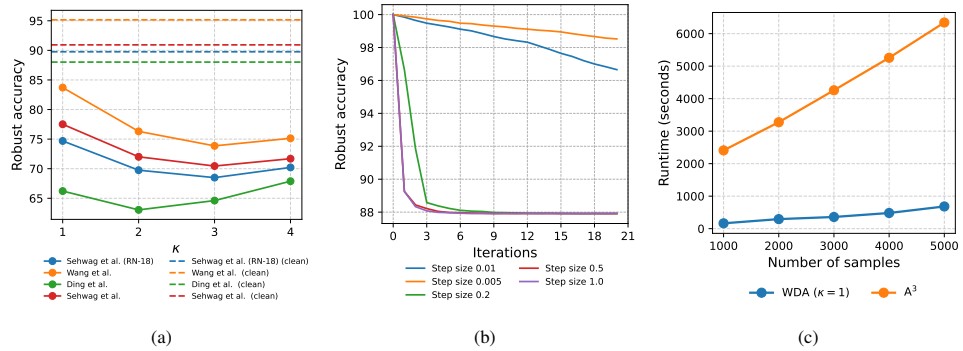

(a)             (b)             (c)

Figure 3: Analysis of $\kappa$ and step size effect. (a) Robust accuracy with varying $\kappa$ on different defense method. (b) Convergence of Wang et al. (2023) under $r_2$ perturbations ($\epsilon = 0.5$). (c) Run time comparision between WDA and $A^3$.

Figure 3a illustrates how the robust accuracy of WDA varies as the parameter $\kappa$ increases. Across all models, raising $\kappa$ beyond 1 generally leads to a noticeable drop in robust accuracy. Specifically, for the models from Wang et al. (2023), Sehwag et al. (2022), and Sehwag et al. (2022) (RN-18), the best performance is observed at $\kappa = 3$, whereas for Ding et al. (2020), the lowest robust accuracy occurs at $\kappa = 2$. These results indicate that increasing $\kappa$ consistently weakens model robustness, with the precise $\kappa$ that produces the largest drop depending on the architecture. Figure 3b presents robust accuracy for the Wang et al. (2023) model with $r_2$ perturbations and with different step sizes for attack. The x-axis represents iterations, and the y-axis shows robust accuracy. In Figure 3b, smaller step sizes (0.01 and 0.005) lead to higher robust accuracy (97%–99%), reflecting weaker attacks. The most effective attack occurs at step size 0.2, where the accuracy drops to around 88%. At larger step sizes (0.5 and 1.0), robust accuracy stabilizes at lower values despite initial drops, suggesting reduced attack effectiveness. Figure 3c reports the runtimes (in seconds) of our WDA and $A^3$ on ImageNet dataset with number of sample ranging from 1000 to 5000. We can observe that WDA has lower computational time and grows slower than $A^3$, verifying the scalability of our method.

## 7 CONCLUSIONS

We presented tight robustness certificates and stronger adversarial attacks for deep neural networks by exploiting their local geometric structure. For ReLU networks, we derived exact WDRO bounds

using their piecewise-affine property, computing data-dependent Lipschitz constants from activation patterns that significantly tighten existing global bounds. For networks with smooth activations (GELU, SiLU), we characterized the worst-case loss through asymptotic Jacobian behavior along adversarial rays, providing the first tractable WDRO analysis for these modern architectures. Our Wasserstein Distributional Attack (WDA) algorithm constructs adversarial distributions on $2N$ points rather than restricting to $N$ perturbed points, achieving lower robust accuracy than state-of-the-art methods across CIFAR-10/100 benchmarks. While WDA incurs additional computational overhead compared to single-point attacks due to evaluating multiple candidate perturbations per iteration, it demonstrates that existing robustness evaluations significantly underestimate vulnerability by considering only point-wise perturbations. Together, these contributions narrow the gap between theoretical certificates and practical evaluation, revealing that both tighter bounds and stronger attacks emerge from properly leveraging network geometry and distributional perspectives.

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

# A  PRELIMINARIES ON WASSERSTEIN DISTANCE AND WDRO

Recall that given two probability measures $\mathbb{P}$ and $\mathbb{Q}$ on $\mathcal{Z}$, the Wasserstein distance is defined as

$$\mathcal{W}_{d,p}(\mathbb{P}, \mathbb{Q}) \triangleq \left( \inf_{\pi \in \Pi(\mathbb{P}, \mathbb{Q})} \int_{\mathcal{Z} \times \mathcal{Z}} d^p(z', z) \, \mathrm{d}\pi(z', z) \right)^{1/p}$$

for $p \in [1, \infty)$, and $\mathcal{W}_{d,p}(\mathbb{P}, \mathbb{Q}) \triangleq \inf_{\pi \in \Pi(\mathbb{P}, \mathbb{Q})} \mathrm{ess.\,sup}_\pi(d)$ for $p = \infty$, where the feasible set is given by

$$\Pi(\mathbb{P}, \mathbb{Q}) \triangleq \left\{ \pi \text{ on } \mathcal{Z} \times \mathcal{Z} : \pi(A \times \mathcal{Z}) = \mathbb{P}(A), \ \pi(\mathcal{Z} \times B) = \mathbb{Q}(B) \ \forall A, B \subseteq \mathcal{Z} \right\}$$

the set of couplings (transport plans) between $\mathbb{P}$ and $\mathbb{Q}$. Intuitively, a transportation plan $\pi$ is feasible if it is a joint distribution whose first marginal is $\mathbb{P}$ and second marginal $\mathbb{Q}$. In the ambiguity set $\Omega_p = \{\mathbb{P} \mid \mathcal{W}_{d,p}(\mathbb{P}, \mathbb{P}_N) \leq \epsilon\}$ (equation 3), a (distributional) attack $\mathbb{P}$ is admissible if the minimal effort for moving mass from $\mathbb{P}$ to the empirical distribution $\mathbb{P}_N$ is not exceeding budget $\epsilon$. Unlike traditional approaches which only allows point-wise perturbations, WDRO min-max model (equation 3) allows both discrete and continuous distribution $\mathbb{P}$, which is extremely practical in certain scenarios where the ground-truth distribution $\mathbb{P}_{\mathrm{true}}$ is unknown and possibly continuous.

# B  PROOFS

## B.1  PROOF OF THEOREM 3.1

**Proof of Upper Bound**  It is a standard result that for any $y = e_k$, if $\ell$ is the cross-entropy loss then

$$\begin{aligned} |\ell(x', y; \theta) - \ell(x, y; \theta)| &= |\log [\mathrm{softmax}\,\theta(x')]_k - \log [\mathrm{softmax}\,\theta(x)]_k| \\ &\leq 2^{1/s} \|\theta(x') - \theta(x)\|_s, \end{aligned} \tag{11}$$

or if $\ell$ is the DLR loss then

$$\begin{aligned} |\ell(x', y; \theta) - \ell(x, y; \theta)| &= |(\max_{k_1 \neq k} \theta(x')_{k_1} - \theta(x')_k) - (\max_{k_2 \neq k} \theta(x)_{k_2} - \theta(x)_k)| \\ &\leq 2^{1/s} \|\theta(x') - \theta(x)\|_s. \end{aligned} \tag{12}$$

In addition, by Jordan & Dimakis (2020), we have that for any $x', x \in \mathcal{X}$,

$$\|\theta(x') - \theta(x)\|_s \leq \max_{\boldsymbol{D} \in \mathcal{D}_\mathcal{X}} \|J_{\boldsymbol{D}}\|_{r \to s} \times \|x' - x\|_r. \tag{13}$$

Thus,

$$\begin{aligned} |\ell(x', y; \theta) - \ell(x, y; \theta)| &\leq 2^{1/s} \max_{\boldsymbol{D} \in \mathcal{D}_\mathcal{X}} \|J_{\boldsymbol{D}}\|_{r \to s} \times \|x' - x\|_r \\ &= \boldsymbol{L} \times d((x', y), (x, y)). \end{aligned} \tag{14}$$

for any $x', x \in \mathcal{X}$ and therefore by using Lipschitz certificate (Mohajerin Esfahani & Kuhn, 2018; Blanchet et al., 2019; Gao & Kleywegt, 2023; Gao et al., 2024; Chu et al., 2024), we have

$$\sup_{\mathbb{P} \,:\, \mathcal{W}_{d,1}(\mathbb{P}, \mathbb{P}_N) \leq \epsilon} \mathbb{E}_{\mathbb{P}}[\ell(Z; \theta)] \leq \mathbb{E}_{\mathbb{P}_N}[\ell(Z; \theta)] + \boldsymbol{L}\epsilon, \tag{15}$$

for any $\epsilon > 0$.

**Proof of Lower Bound**  To show that the lower bound of the worst-case loss is $\mathbb{E}_{\mathbb{P}_N}[\ell(Z; \theta)] + \boldsymbol{l}\epsilon$, it is enough to construct a perturbation $\tilde{Z}$, a weight $\eta \in (0, 1]$, and a distribution

$$\mathbb{P}_{\mathrm{adv}} = \sum_{i=1, i \neq \iota}^{N} \frac{1}{N} \boldsymbol{\delta}_{Z^{(i)}} + \frac{1 - \eta}{N} \boldsymbol{\delta}_{Z^{(\iota)}} + \frac{\eta}{N} \boldsymbol{\delta}_{\tilde{Z}}, \tag{16}$$

so that $\mathcal{W}_{d,1}(\mathbb{P}_{\mathrm{adv}}, \mathbb{P}_N) \leq \epsilon$ and $\mathbb{E}_{\mathbb{P}_{\mathrm{adv}}}[\ell(Z; \theta)] \approx \mathbb{E}_{\mathbb{P}_N}[\ell(Z; \theta)] + \boldsymbol{l}\epsilon$.

Since $\mathcal{D}_\mathcal{X}$ is finite, let $x^\star, \boldsymbol{D}^\star, k'^\star, k^\star$ and sequence $\{u_t^\star\}$ be the maximizer in 7, i.e., $\boldsymbol{D}^\star \in \mathcal{D}_{x^\star}, k'^\star \neq k^\star, \{u_t^\star\} \subset \mathrm{int}(\mathrm{rec}(\mathcal{C}_{\boldsymbol{D}^\star}))$ and

$$(e_{k'^\star} - e_{k^\star})^\top J_{\boldsymbol{D}^\star} u_t^\star \to \boldsymbol{l} \text{ when } t \to \infty.$$

In particular, $\theta$ is affine and differentiable on $\mathrm{rec}(\mathcal{C}_{\mathbf{D}^\star})$. Since $u_t^\star$ belongs to the open cone $\mathrm{rec}(\mathcal{C}_{\mathbf{D}^\star})$, one has that for any $\alpha > 0$,

$$\tilde{x} = x^\star + \alpha u_t^\star \in \mathrm{rec}(\mathcal{C}_{\mathbf{D}^\star}), \tag{17}$$

and thus

$$\nabla_x \theta(\tilde{x}) = J_{\mathbf{D}^\star}, \quad \theta(\tilde{x}) - \theta(x^\star) = \alpha J_{\mathbf{D}^\star} u_t^\star. \tag{18}$$

Choose root $\iota$ so that $(X^{(\iota)}, Y^{(\iota)} = \mathbf{e}_{k^\star})$. Then when $\ell$ is the cross-entropy loss or DLR loss, by a technical Lemma B.1 one has

$$\lim_{\alpha \to \infty} \frac{\Delta \ell(\alpha)}{\alpha} = \lim_{\alpha \to \infty} \frac{\ell(\tilde{x}, Y^{(\iota)}; \theta) - \ell(x^\star, Y^{(\iota)}; \theta)}{\alpha} \geq v_{k'^\star} - v_{k^\star}. \tag{19}$$

where $v = J_{\mathbf{D}^\star} u_t^\star$. Now choose $\alpha$ large enough so that $\Delta \ell(\alpha) \approx \alpha(v_{k'^\star} - v_{k^\star})$, $\Delta \ell(\alpha) \gg \ell(x^\star, Y^{(\iota)}; \theta) - \ell(X^{(\iota)}, Y^{(\iota)}; \theta)$, and $N\epsilon < \|\tilde{x} - X^{(i)}\|_r \approx \alpha$. Set $\tilde{Z} = (\tilde{x}, Y^{(\iota)})$, then

$$\begin{aligned} \ell(\tilde{Z}; \theta) - \ell(Z^{(\iota)}; \theta) &= \Delta \ell(\alpha) + \ell(x^\star, Y^{(\iota)}; \theta) - \ell(X^{(\iota)}, Y^{(\iota)}; \theta) \\ &\approx \|\tilde{x} - X^{(i)}\|_r (v_{k'^\star} - v_{k^\star}) \\ &= d(\tilde{Z}, Z^{(\iota)}) \times (\mathbf{e}_{k'^\star} - \mathbf{e}_{k^\star})^\top J_{\mathbf{D}^\star} u_t^\star \xrightarrow{t \to \infty} \mathbf{l} \times d(\tilde{Z}, Z^{(\iota)}). \end{aligned} \tag{20}$$

Now set $\eta = \frac{N\epsilon}{d(\tilde{Z}, Z^{(\iota)})} \in (0, 1]$, then

$$\mathcal{W}_{d,1}(\mathbb{P}_{\mathrm{adv}}, \mathbb{P}_N) \leq \frac{\eta}{N} d(\tilde{Z}, Z^{(\iota)}) = \epsilon. \tag{21}$$

Moreover,

$$\begin{aligned} \mathbb{E}_{\mathbb{P}_{\mathrm{adv}}}[\ell(Z; \theta)] &= \mathbb{E}_{\mathbb{P}_N}[\ell(Z; \theta)] + \frac{\eta}{N}\left(\ell(\tilde{Z}; \theta) - \ell(Z^{(\iota)}; \theta)\right) \\ &\approx \mathbb{E}_{\mathbb{P}_N}[\ell(Z; \theta)] + \frac{\epsilon}{d(\tilde{Z}, Z^{(\iota)})} \mathbf{l} d(\tilde{Z}, Z^{(\iota)}) \\ &= \mathbb{E}_{\mathbb{P}_N}[\ell(Z; \theta)] + \mathbf{l}\epsilon. \end{aligned} \tag{22}$$

Therefore, the lower bound of the worst-case loss is $\mathbb{E}_{\mathbb{P}_N}[\ell(Z; \theta)] + \mathbf{l}\epsilon$.

**Sufficient condition of $\mathbf{l} = \mathbf{L}$** Suppose that the dual-norm maximizer $\xi = \mathcal{M}_r(J_{\mathbf{D}^\star}^\top(\mathbf{e}_{k'^\star} - \mathbf{e}_{k^\star})) \in \mathrm{rec}(\mathcal{C}_{\mathbf{D}^\star})$ where $\mathbf{D}^\star$ is a maximizer of 6 and $(k'^\star, k^\star)$ is a maximizer of 7, then we have

$$\begin{aligned} \mathbf{l} &= (\mathbf{e}_{k'^\star} - \mathbf{e}_{k^\star})^\top J_{\mathbf{D}^\star} u_t^\star \\ &\geq (\mathbf{e}_{k'^\star} - \mathbf{e}_{k^\star})^\top J_{\mathbf{D}^\star} \xi && \text{(since } u_t^\star \text{ is the maximizer)} \\ &= \|(\mathbf{e}_{k'^\star} - \mathbf{e}_{k^\star})^\top J_{\mathbf{D}^\star}\|_s && \text{(by definition of dual-norm maximizer)} \\ &= \|(\mathbf{e}_{k'^\star} - \mathbf{e}_{k^\star})\|_r \|J_{\mathbf{D}^\star}\|_{r \to s} \\ &= 2^{1/s} \|J_{\mathbf{D}^\star}\|_{r \to s} = \mathbf{L}, \end{aligned} \tag{23}$$

where the second last equality holds true because $(\mathbf{e}_{k'^\star} - \mathbf{e}_{k^\star})$ is the largest increment direction of $J_{\mathbf{D}^\star}$.

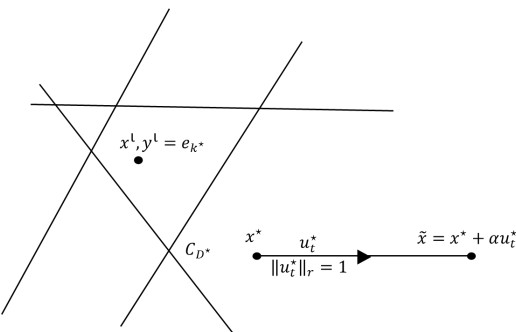

Figure 4: Illustration of Proof of Lower Bound

## B.2 PROOF OF THEOREM 3.3

*Proof.* The proof for the upper and lower bounds is similar to the methodology we discussed in our previous exchange.

**Proof of Upper Bound** The WDRO upper bound is a direct consequence of the Lipschitz continuity of the loss function. The Lipschitz constant of the combined loss function, $L_\ell = \sup_{Z \in \mathcal{Z}} \|\nabla_x \ell(Z; \theta)\|_r$, is bounded by the product of the Lipschitz constant of the loss with respect to the output and the Lipschitz constant of the network. That is,

$$\boldsymbol{L} \le \|J(x)\|_{r \to s} \cdot \|\nabla_\theta \ell\|_s = \|J(x)\|_{r \to s} \cdot 2^{1/s}$$

**Proof of Lower Bound** The proof of the lower bound is identical with the ReLU network case, where it relies on constructing a specific adversarial distribution. This finds a point $x^\star$ and a direction $u^\star$ that maximize the rate of change of the loss. The constant $\boldsymbol{l}$ is defined as this maximum rate of change. By constructing a perturbed point $\tilde{x} = x^\star + \alpha u^\star$ and a corresponding adversarial distribution, it is shown that the worst-case loss is at least $\mathbb{E}_{\mathbb{P}_N}[\ell(Z; \theta)] + \boldsymbol{l}\epsilon$.

$\square$

## B.3 TECHNICAL PROOFS

**Lemma B.1** (Technical lemma). *In equation 19, if $\ell = \ell_{CE}$ is the cross-entropy loss, then*

$$\lim_{\alpha \to \infty} \frac{\Delta \ell_{CE}}{\alpha} = \max_i (J_{\boldsymbol{D}^\star} u_t^\star)_i - (J_{\boldsymbol{D}^\star} u_t^\star)_{k^\star}.$$

*Else if $\ell = \ell_{DLR}$ is the DLR loss, then*

$$\lim_{\alpha \to \infty} \frac{\Delta \ell_{DLR}}{\alpha} = \max_{i \ne k^\star} (J_{\boldsymbol{D}^\star} u_t^\star)_i - (J_{\boldsymbol{D}^\star} u_t^\star)_{k^\star}.$$

*Proof.* Let $\theta^\star = \theta(x^\star)$ and the change in network output be $\Delta\theta = \theta(\tilde{x}) - \theta(x^\star) = \alpha J_{\boldsymbol{D}^\star} u_t^\star$. We will analyze the limit for each loss function separately.

**Cross-Entropy Loss:** The difference in loss is $\Delta \ell_{CE} = \ell_{CE}(\theta(\tilde{x}), e_{k^\star}) - \ell_{CE}(\theta(x^\star), e_{k^\star})$. Using the property $\ell_{CE}(z, e_{k^\star}) = -(z_{k^\star} - \log \sum_k e^{z_k})$, the loss difference is:

$$\Delta \ell_{CE} = -\Delta\theta_{k^\star} + \log\left(\sum_k e^{\Delta\theta_k} \cdot \text{softmax}(\theta^\star)_k\right)$$

To find the limit of the average rate of change, $\frac{\Delta \ell_{CE}}{\alpha}$, we substitute $\Delta\theta = \alpha v$, where $v_k = (J_{\boldsymbol{D}^\star} u_t^\star)_k$.

$$\lim_{\alpha \to \infty} \frac{\Delta \ell_{CE}}{\alpha} = \lim_{\alpha \to \infty} \left[\frac{1}{\alpha} \log\left(\sum_k \text{softmax}(\theta^\star)_k e^{\alpha v_k}\right) - v_{k^\star}\right]$$

Let $v_{\max} = \max_k v_k$. Factoring out the dominant term $e^{\alpha v_{\max}}$ from the sum, the expression becomes:

$$= \lim_{\alpha \to \infty} \left[\frac{1}{\alpha}\left(\log(e^{\alpha v_{\max}}) + \log\left(\sum_k \text{softmax}(\theta^\star)_k e^{\alpha(v_k - v_{\max})}\right)\right) - v_{k^\star}\right]$$

$$= \lim_{\alpha \to \infty} \left[v_{\max} + \frac{1}{\alpha} \log\left(\sum_k \text{softmax}(\theta^\star)_k e^{\alpha(v_k - v_{\max})}\right) - v_{k^\star}\right]$$

The sum inside the logarithm converges to a constant value, as all terms with $v_k < v_{\max}$ go to 0. The logarithmic term is therefore bounded. The term $\frac{1}{\alpha}$ causes the entire second term to go to 0. The limit is thus:

$$= v_{\max} - v_{k^\star} = \max_k (J_{\boldsymbol{D}^\star} u_t^\star)_k - (J_{\boldsymbol{D}^\star} u_t^\star)_{k^\star}$$

**DLR Loss:** The difference in DLR loss is $\Delta \ell_{DLR} = \ell_{DLR}(\theta(\tilde{x}), k^\star) - \ell_{DLR}(\theta(x^\star), k^\star)$.

$$\Delta \ell_{DLR} = \left(\max_{k \ne k^\star} \theta(\tilde{x})_k - \theta(\tilde{x})_{k^\star}\right) - \left(\max_{k \ne k^\star} \theta(x^\star)_k - \theta(x^\star)_{k^\star}\right)$$

Substituting $\theta(\tilde{x}) = \theta^\star + \Delta\theta$:

$$\Delta\ell_{DLR} = \left(\max_{k \neq k^\star}(\theta_k^\star + \Delta\theta_k) - \max_{k \neq k^\star}\theta_k^\star\right) - \Delta\theta_{k^\star}$$

To find the limit of the average rate of change, $\frac{\Delta\ell_{DLR}}{\alpha}$, we substitute $\Delta\theta = \alpha v$ and analyze as $\alpha \to \infty$.

$$\lim_{\alpha \to \infty} \frac{\Delta\ell_{DLR}}{\alpha} = \lim_{\alpha \to \infty} \frac{1}{\alpha}\left(\max_{k \neq k^\star}(\theta_k^\star + \alpha v_k) - \max_{k \neq k^\star}\theta_k^\star\right) - v_{k^\star}$$

As $\alpha \to \infty$, the term $\alpha v_k$ dominates inside the maximum function. The limit of the maximum term is therefore $\max_{k \neq k^\star} v_k$.

$$\lim_{\alpha \to \infty} \frac{\Delta\ell_{DLR}}{\alpha} = \left(\max_{k \neq k^\star} v_k\right) - v_{k^\star} = \max_{k \neq k^\star}(J_{\boldsymbol{D}^\star}u_t^\star)_k - (J_{\boldsymbol{D}^\star}u_t^\star)_{k^\star}$$

$\square$

## C  ADDITIONAL RESULT

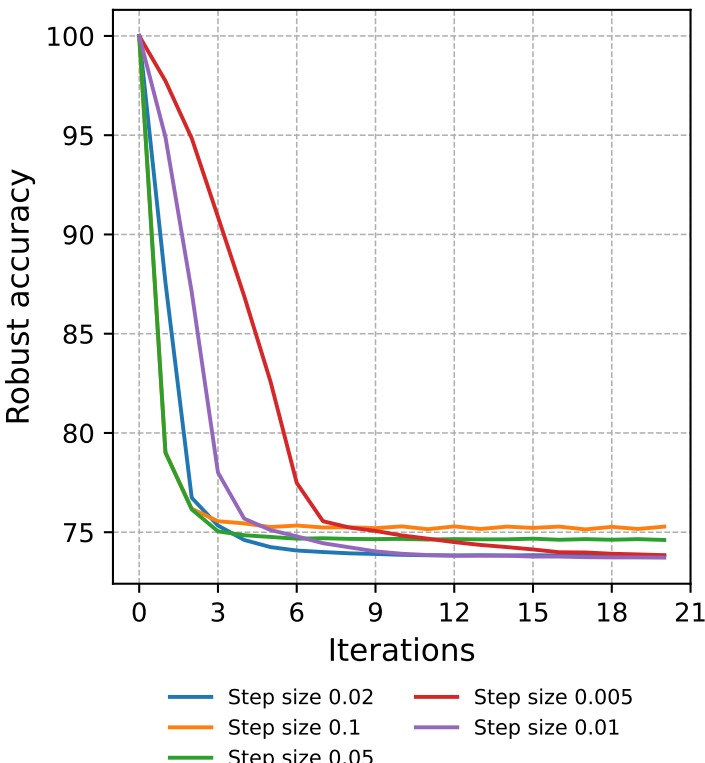

Figure 5: Comparison of step size effect on model Cui et al. (2024) under $r_\infty$ perturbation ($\epsilon = 8/255$).

Figure 5 illustrates the convergence behavior of Cui et al. (2024) in terms of robust accuracy. Larger step sizes (0.1, 0.5) lead to higher final accuracy, whereas a smaller step size of 0.02 results in the lowest robust accuracy, indicating the most effective attack.

Figure 6 presents the robust accuracy across different values of $\kappa$. Among the tested settings, $\kappa = 2$ consistently produces the lowest robust accuracy for all models (Cui et al. (2024), Wang et al. (2023), and Sehwag et al. (2022)).

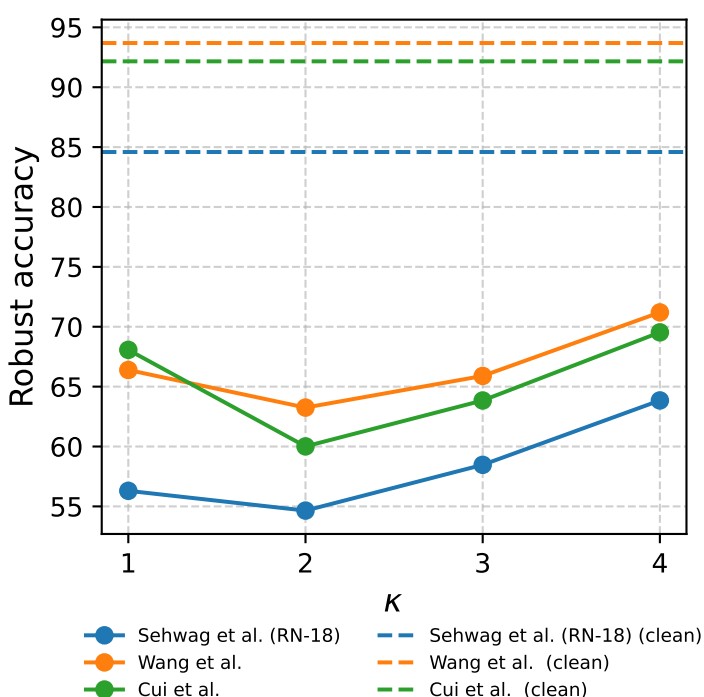

Figure 6: Comparison of robust accuracy with varying $\kappa$

## D    USAGE OF LARGE LANGUAGE MODELS (LLMS)

We used ChatGPT solely for revising the writing of the paper. Note that revision here strictly means enhancing the clarity and readability of the text (e.g., fixing typos or constructing latex tables), and not for any other purposes.

