# OpenReview forum: "Tight Robustness Certificates and Wasserstein Distributional Attacks for Deep Neural Networks"
_ICLR.cc/2026/Conference — Submitted to ICLR 2026_

### Official Review · Reviewer_t19d · 2025-10-29

**Soundness:** 3
**Presentation:** 3
**Contribution:** 2
**Rating:** 4
**Confidence:** 4

**Summary:**

This work proposes a new type of adversarial attack along with an alternative approach for more precisely measuring model robustness.

**Strengths:**

Unlike existing adversarial attacks, the proposed method flexibly generates distributional adversarial examples, which could offer a different perspective on robustness evaluation.

**Weaknesses:**

The proposed attack does not appear to outperform existing methods when κ (kappa) is set to 1, and when transformer-based models are included in the experiments. This limitation diminishes the overall contribution of the work.

Specifically, the main purpose of AutoAttack (AA) is to reduce the influence of manually chosen hyperparameters, though it requires significant computational time to complete. The proposed Adaptive Auto Attack (A³ attack) aims to find competitive adversarial examples more efficiently. However, the generation time for adversarial examples has not been reported. Additionally, as shown in Figure 3, the proposed attack seems to struggle with selecting an appropriate step size. Together, these issues make it difficult to clearly assess the contribution of this work.

**Questions:**

* **Role of ReLU:** ReLU appears to play a central role in this method. I wonder whether the proposed attack can be extended to transformer-based models or to networks using other activation functions. If not, this limitation significantly reduces the generalizability and impact of the proposed method, especially compared with existing attacks. If it can be extended, please provide empirical evidence.

* **Jacobian Requirement:** If I understand correctly, the proposed attack requires computation of the Jacobian, which implicitly means it cannot be applied in black-box settings. Please clarify whether this limitation applies.

* **Target Selection (Algorithm 1):** The authors claim that “WDA consistently outperforms other single-method attacks” (lines 430–431). However, in Algorithm 1 (line 11), the target j* is altered in each iteration. Does this mechanism effectively play a role similar to that of ensemble attacks?

---

> ### Author Response · Authors · 2025-11-21
> **Response to Reviewer t19d**
>
> We thank the reviewer for the careful reading and for highlighting both the strengths and limitations of our work. We address the concerns below.
> ## 1. Additional detail about architecture
> ### 1.1 ReLU is one case of WDRO analysis and we have general analysis for smooth activation function
> Our theoretical development already distinguishes two activation regimes ReLU networks (Section 3.1, Theorem 3.1) and Smooth activations (Section 3.2, Theorem 3.3). The WDA algorithm (Algorithm 1) itself is agnostic to the activation: it only requires gradients of logit differences. We believe part of the confusion stems from the fact that the experiments may have looked “ReLU-like” by architecture names. In fact, the RobustBench WideResNet models we use in our experiments employ SiLU activations, not ReLU. These models are therefore directly covered by the smooth-activation regime of Section 3.2, not by the ReLU-specific Section 3.1. This already provides empirical evidence that WDA is effective for non-ReLU activations.
> ### 1.2. About transformer-based models
> We would like to clarify that our work only reported robustness experiments on convolutional-based models (ResNet and WideResNet) and transformer-based models were not included.
> ### 1.3. Exact tractable reformulation of WDRO is impossible in certain black-box settings
> We thank the reviewer for raising this concern. Indeed, for certain black-box settings, it is impossible to obtain a tractable reformulation of WDRO exactly, even in the case of knowing the exact Lipschitz constant. For instance, consider the binary CE loss $\ell_1(x)= \log(1+\exp(x))$ and the (hard) sigmoid loss $\ell_2(x)= \max(0, \min(1, (1-x)/2))$: both of them are 1-Lipschitz, but their exact interpretations are completely different, see [1, 2, 3, 4, 5]. Since theoretical analysis is not applied, our WDA is not available. We note that all of other baselines used in this work are white-box attacks (APGD, AA, $A^3$).
> ## 2. Additional detail of WDA
> ### 2.1. WDA ($\kappa=1$) reduces to point-wise attack, WDA ($\kappa > 1$) is a distributional attack and our contribution
> We thank the reviewer for raising the issue of performance at $\kappa=1$. We would like to emphasize that WDA is a generalization of existing methods and WDA with $\kappa = 1$ is a special case where our WDA reduces to a point-wise attack, see Figure 1. Therefore, it's not expected that WDA ($\kappa = 1$) outperform other point-wise attacks (APGD, AA, $A^3$).
> ### 2.2. Additional computational time experiment
> We appreciate the reviewer’s emphasis on computational cost and hyperparameter sensitivity. We have revised the paper accordingly and added the computational time into the ablation study.
> | Number of ImageNet samples | 1000 | 2000 | 3000 | 4000 | 5000 |
> |---------------------------|------|------|------|------|------|
> | WDA ($\kappa=1$)|164.34|293.62|359.3|482.02|683.09|
> | $A^3$|2408.54|3275.61|4258.15|5257.92|6342.29|
>
> This reports the runtimes (in seconds) of our WDA and $A^3$ on ImageNet dataset with number of sample ranging from 1000 to 5000. We can observe that WDA has lower computational time and grows slower than $A^3$, verifying the scalability of our method.
> ### 2.3. Step size is fixed in our experiments
> Figure 3b is an ablation study of choosing step size: the x-axis sweeps different fixed step sizes $\alpha$ to illustrate how robust accuracy changes. From this figure, we choose the step size to be 0.2 for all experiments involving $r_2=0.5$. Similarly, we choose $\alpha\in\{0.01, 0.02\}$ for experiments involving $r_{\infty}\in\{4/255, 8/255\}$.
> ### 2.4. Target selection in Algorithm 1 and relation to ensemble attacks
> We would like to clarify the difference between our target selection in Algorithm 1 and ensemble attack mechanism:
> - At every iteration, our WDA considers all classes, select the optimal class $j^*$ to perturb the current sample, and discard the remaining candidates.
> - On the other hand, ensemble attacks like AA sequentially execute four single attack methods and the cross-cluster interaction only happen when ensemble method switches from one method to another. Whereas, $A^3$ strategically discards hard-to-attack samples at a certain rate.
>
> In conclusion, our WDA is not an ensemble attack.
>
> *We thank the reviewer again for taking their time and effort to provide us with valuable feedback, and we are eager to address any further concerns from the reviewers.*

---

> > ### Author Response · Authors · 2025-11-21
> > **References**
> >
> > [1] Mohajerin Esfahani, et al. "Data-driven distributionally robust optimization using the Wasserstein metric: Performance guarantees and tractable reformulations." Mathematical Programming 2018
> >
> > [2] Blanchet, Jose, et al. "Robust Wasserstein profile inference and applications to machine learning." Journal of Applied Probability 2019
> >
> > [3] Gao, Rui, et al. "Distributionally robust stochastic optimization with Wasserstein distance." Mathematics of Operations Research 2023
> >
> > [4] Gao, Rui, et al. "Wasserstein distributionally robust optimization and variation regularization." Operations Research 2024
> >
> > [5] Chu, Hong, et al. "Wasserstein distributionally robust optimization and its tractable regularization formulations." arXiv 2024

---

### Official Review · Reviewer_ckur · 2025-10-31

**Soundness:** 4
**Presentation:** 4
**Contribution:** 4
**Rating:** 8
**Confidence:** 3

**Summary:**

The paper extends standard point-wise robustness evaluation to a distributional setting by introducing a Wasserstein Distributional Attack (WDA) parameterized by a parameter $\kappa$, which induces a strictly harder threat model. The paper further derives both upper and lower bounds on the worst-case loss, and attempts to give the sufficient conditions under which these bounds become tight. On the empirical side, this work further instantiates a concrete attack algorithm and shows that increasing the parameter kappa consistently reduces robust accuracy on CIFAR-10/100 under, thereby confirming that the theoretical bounds are consistent with practical robustness. Overall, the paper makes a practical contribution by pairing a stronger distributional threat model with bounds that are actually computable and reasonably tight.

**Strengths:**

One of the paper’s key strengths is how cleanly it upgrades point-wise robustness evaluation to distributional robustness with one intuitive $\kappa$ parameter, which naturally includes prior point-wise cases as its special case.

In addition, the paper provides a concrete Wasserstein Distributional Attack and gives computable risk bounds, which turns WDRO into a practical tool. For ReLU networks, the activation-mask/Jacobian treatment even yields constructive (sometimes tight) certificates. The evaluation results further supports this claim: tuning the parameter $\kappa$ effectively controls the robust accuracy.

Taken together, the reviewer believes this work validates WDRO as a workable notion of robustness, and makes a strong case that future robustness certificates should target this distributional shifts rather than prior point-wise evaluation, and that defenses/training objectives should minimize this kind of distributional risk as in the paper.

**Weaknesses:**

The paper cites rLR-QP and BaB-Attack, but they don’t appear in the experiments. Is there a reason for omitting them (e.g., scalability or implementation constraints)? Since these are point-wise attacks—--i.e., the kappa=1 special case of your proposed WDA---it would be helpful to include them as baselines in the point-wise setting or clarify why they’re not considered.

Another minor typo: in the metrics description, the distributional accuracy is written as (1−1/kappa)⋅clean + kappa⋅adversarial, but given your mixture weights in Alg. 1/Eq. (10), the adversarial term should be 1/kappa instead of kappa.

**Questions:**

Please include rLR-QP and BaB-Attack as baselines in the evaluation or briefly explain why they’re not included.

---

> ### Author Response · Authors · 2025-11-21
> **Response to Reviewer ckur**
>
> We thank the reviewer for the very detailed and positive assessment of our work, and for the concrete suggestions that help strengthen the paper.
>
> ## 1. On rLR-QP and BaB-Attack as $\kappa=1$ baselines
> We appreciate the suggestion to include rLR-QP [1] and BaB-Attack [2] as $κ=1$ baselines, and agree they are important contributions to the adversarial-attack literature. However, after careful consideration we decided not to adopt them as default baselines in our setting, for the following reasons.
>
> - **Both methods are designed specifically for ReLU networks**. rLR-QP explicitly targets its derivation and experiments for fully connected ReLU architectures. BaB-Attack also operates on ReLU networks by formulating adversarial search as a branch-and-bound procedure over activation patterns in a mixed-integer formulation. In contrast, our evaluation includes modern convolutional models with smooth activations (e.g., SiLU).
> - **Both attacks are considerably more expensive than APGD**: As pointed out in [4], rLR-QP encounter scalability issue and is only applied to small networks. Additionally, BaB-Attack is significantly slower than gradient based method which make them unsuitable for attacking larger model [2].
>
> For these reasons, we opted not to add rLR-QP and BaB-Attack as routine baselines since we already include more efficient APGD [3]. We highlight this discussion in the revised manuscript.
> > As pointed out in Zhang et al. (2022); Croce et al. (2020), rLR-QP and BaB-Attack are not as efficient as gradient based attack, therefore, we only use APGD as single-method baseline in our experiment.
>
> ## 2. Typo in notation of $\kappa$
> Thank you for addressing the typo in the metric description, we have corrected the formula and highlighted in the revision (Section 6, lines 428–429). The implementation and reported numbers already used the correct weights and the experimental results do not change.
>
> *We thank the reviewer again for taking their time and effort to provide us with valuable feedback, and we are eager to address any further concerns from the reviewers.*
>
> ---
> [1] Croce, Francesco and Matthias Hein. "A randomized gradient-free attack on relu networks." GCPR 2018
>
> [2] Zhang, Huan, et al. "A branch and bound framework for stronger adversarial attacks of relu networks." ICML 2022
>
> [3] Croce, et al. "Reliable evaluation of adversarial robustness with an ensemble of diverse parameter-free attacks." ICML 2020
>
> [4] Croce, et al. "Scaling up the randomized gradient-free adversarial attack reveals overestimation of robustness using established attacks." IJCV 2020

---

### Official Review · Reviewer_p2JU · 2025-11-03

**Soundness:** 3
**Presentation:** 3
**Contribution:** 3
**Rating:** 6
**Confidence:** 3

**Summary:**

The paper proposes a new framework for distributional adversarial robustness under Wasserstein threat models while also providing theory along with it. More specifically regarding theory, it proposes a tight Lipschitz-based robustness certificates which are derived from the exact local geometry of ReLU and smooth neural networks. And regarding the experimental part, it proposes a novel Wasserstein Distributional Attack that generates distributional rather than point-wise adversarial examples.

**Strengths:**

- The paper has a good combination of both theory and experiments with many comparisons with other methods.
- Good presentation of the method and the experiments, and also an extensive ablation study.

**Weaknesses:**

- The paper is kind of hard to follow and read if you don't have experience with the Wasserstein distribution. But apart from that everything is well explained and presented.

**Questions:**

- I do think it is a good paper for the venue and fits the usual ICLR framework. Some questions that I would ask is how do the authors compute the step for every attack step?
- Regarding scalability can you provide attack runtimes and how it scales on bigger models that you are using here?

---

> ### Author Response · Authors · 2025-11-21
> **Response to Reviewer p2JU**
>
> We thank the reviewer for the positive assessment of our theoretical and numerical results. We would like to address the reviewer's questions as follow:
>
> ## 1. Step size is fixed in our experiments
> Figure 3b is an ablation study of choosing step size: the x-axis sweeps different fixed step sizes $\alpha$ to illustrate how robust accuracy changes. From this figure, we choose the step size $\alpha=0.2$ for all experiments involving $r_2=0.5$. Similarly, we choose the step size $\alpha\in\{0.01, 0.02\}$ for budget $r_{\infty}\in\{4/255, 8/255\}$ respectively.
>
> ## 2. Runtime and scalability
> We would like to address the scalability of our method by adding the runtime experiment in the following table. (See Figure 3c in our revised manuscript.)
>
> | Method / Number of sample | 1000 | 2000 | 3000 | 4000 | 5000 |
> |---------------------------|------|------|------|------|------|
> | WDA ($\kappa=1$)|164.34|293.62|359.3|482.02|683.09|
> | $A^3$|2408.54|3275.61|4258.15|5257.92|6342.29|
>
> This reports the runtimes (in seconds) of our WDA and $A^3$ on ImageNet dataset with number of sample ranging from 1000 to 5000. We can observe that WDA has lower computational time and grows slower than $A^3$, verifying the scalability of our method.
>
> ## 3. Additional preliminaries on WDRO
> We appreciate the reviewer's feedback on the presentation of our work. In the revision, we added reference to WDRO literature (lines 138-140) and a notation appendix (Appendix A). These additions aim to improve the paper's accessibility.
>
> *We thank the reviewer again for taking their time and effort to provide us with valuable feedback, and we are eager to address any further concerns from the reviewers.*

---

### Official Review · Reviewer_vVTL · 2025-11-04

**Soundness:** 2
**Presentation:** 2
**Contribution:** 2
**Rating:** 2
**Confidence:** 4

**Summary:**

This paper studies tighter bounds for Wasserstein Distributional Robustness (WDRO). Based on these bounds, it proposes a new adversarial attack.

**Strengths:**

- The bounds for WDR are tighter.

**Weaknesses:**

- It is unclear the role of tighter bounds in developing Wasserstein distributional attack. As far as I see, Theorems 3.1 and 3.3 present to how to estimate two factors $l$ and $L$ relevant to the lower and upper bounds. After Corollary 3.2, the authors claim: "In the spirit of formulation 9, we propose a practical adversarial attack in Section 4 which aims to find adversarial direction $u^{(i)}$ for each sample i so that it maximizes the change of the logit function". Indeed, this is a typical and well-known way to craft adversarial examples without the theories from this paper.
- Although the paper aims to construct adversarial distribution (i.e., attack over distribution), in its Algorithm 1, it runs point-wise attack for each data example $X^{(i)}$. I cannot see any idea of distributional attack here because this requires the interaction of data examples during attack.  Moreover, it is unclear why the authors put clean examples $X^{(i)}$ in the adversarial distribution.
- No experiments on ImageNet which is the common standard for papers in adversarial attacks currently.

**Questions:**

- Why the robust accuracies increase when incorporating WDA to A3 in A3++?
- Wrong order of nested sets in Line 140?

---

> ### Author Response · Authors · 2025-11-21
> **Response to Reviewer vVTL**
>
> We thank Reviewer vVTL for the careful reading and constructive comments. We especially appreciate the Reviewer's recognition that our bounds for WDRO are tighter. Below, we address each concern in turn, and we have revised the paper accordingly.
>
> ## 1. Tighter Bounds Play a Crucial Role in WDA
> We thank the reviewer for raising this concern. We would like to emphasize that the tighter WDRO bounds play a central role in developing our WDA. In particular, the lower bound identifies a candidate worst-case direction for moving probability mass (through the dual variables of the WDRO problem), while the upper bound shows that, under the conditions of Theorems 3.1 and 3.3, this direction is theoretically justified. In certain scenarios where our lower and upper bounds are equal (e.g., Figure 2a), the distribution constructed by **our attack WDA (Eq. (10)) coincides with the theoretical worst-case distribution (Eq. (16)).**
>
> ## 2. The proposed WDA is truly distributional and strictly distinguished from the existing point-wise attack.
> Our WDA is different from existing point-wise attack as follows:
>  - **Number of supports**: Our WDA introduces a different way to craft adversarial examples compared to existing methods as follows. Standard point-wise attacks perturb each input independently and produce a single adversarial example per sample. In contrast, our WDA produces 2 adversarial examples per sample where one of them is the clean sample. Consequently, **WDA constructs a distribution of 2$N$ supports with customized weight**  instead of 1$N$ as in point-wise attack (see lines 324–329).
>  - **Attack direction**: Furthermore, WDA computes attack direction based on our theoretical analysis which compares different candidates and selects optimal direction (Figure 2b). On the other hand, existing PGD-based attack like APGD [3] or DeepFool [4] propose a deterministic direction (and do not consider the cluster information).
>  - **Logits information**: WDA is not an ad-hoc logit-gradient heuristic, but a constructive optimizer of the same WDRO problem that our bounds analyze because when the lower and upper bounds match, the distribution produced by WDA (Eq. 10) is the same as the theoretical worst-case distribution (Eq. 16).
>
> ## 3. Experiments on ImageNet
> In the revised version, we have added ImageNet experiments with the same settings as in RobustBench. The results for ConvNeXt-B from [1] and ConvNeXt-S+ConvStem [2] are shown below:
>
> >| Model | Clean | APGD-CE | APGD-DLR | WDA ($\kappa=1$) | WDA ($\kappa=2$) | AA | $A^3$ | $A^3$++|
> >| - | - | - | - | - | - | - | - | - |
> >| ConvNeXt-B | 76.02 | 55.90 | 56.78 | 54.38 | 52.95 | 55.82 | 53.19 | 53.18 |
> >| ConvNeXt-S+ConvStem | 74.10 | 52.82 | 53.20 | 51.04 | 50.31 | 52.42 | 49.92 | 49.90 |
>
> Similar to CIFAR-10 and CIFAR-100, on ImageNet, WDA($\kappa=1$) achieves robust accuracy lower than APGD and comparable to ensemble-based methods AA and $A^3$. Furthermore, WDA ($\kappa=2$) also produces lower robust accuracy values compare to WDA ($\kappa=1$).
>
> ## 4. Behavior of $A^3$++
> $A^3$ is an ensemble methods comprising of PGD + ADI + ODS (line 382). In our experiments, $A^3$++ is a variant of $A^3$, replacing PGD with our WDA ($\kappa=1$).
>
> The original $A^3$ was carefully tuned for its own PGD-based inner attack, with many hard-coded hyperparameters (step-size schedule, restarts, ADI/ODS thresholds, etc.). In our evaluation, we kept all of these hyperparameters fixed and only swapped PGD for WDA ($\kappa=1$), to make the comparison reproducible and avoid re-tuning the entire framework. However, this also means that the $A^3$ heuristics are not optimized for WDA ($\kappa=1$). As a result, in some configurations our $A^3$++ can be slightly less aversive than the original $A^3$ with PGD, leading to a marginally higher robust accuracy.
>
> We emphasize that this effect is specific to $A^3$ tuned for PGD. Our main conclusions about WDA do not rely on $A^3$++: in its standalone form, WDA consistently reduces robust accuracy compared to APGD, demonstrating its effectiveness.
>
> ## 5. Order of nested sets (Line 138)
> We have corrected the typo.
>
> *We thank the reviewer again for taking their time and effort to provide us with valuable feedback, and we are eager to address any further concerns from the reviewers.*
>
> ---
> [1] Liu, Chang, et al. "A Comprehensive Study on Robustness of Image Classification Models: Benchmarking and Rethinking." IJCV 2025
>
> [2] Singh, Naman Deep, et al. "Revisiting Adversarial Training for ImageNet: Architectures, Training and Generalization across Threat Models." NeurIPS 2023
>
> [3] Croce, et al. "Reliable evaluation of adversarial robustness with an ensemble of diverse parameter-free attacks." ICML 2020
>
> [4] Moosavi-Dezfooli, et al. "Deepfool: a simple and accurate method to fool deep neural networks." CVPR 2016

---

### Author Response · Authors · 2025-11-21
**Overall Response**

We thank all reviewers for their thoughtful and constructive feedback. We have carefully revised the paper to address both conceptual and experimental concerns. Please refer to the updated pdf of the revised submission. Below, we summarize the general concerns from reviewers and our general clarifications.

## 1. Distinguished contribution toward distributional attack and distributional robustness
Concerns were raised about how our theoretical analysis connects with our proposed attack WDA, and whether WDA is different significantly compared to existing attack. In this response, we clarify that:
 - Our tighter theoretical WDRO bounds directly leads to the formulation of WDA: the theoretical worst-case distribution (equation (10)) used in our theoretical proof is the target output of our proposed algorithm  (line 16 of Algorithm 1).
 - WDA is truly distributional and distinct from point-wise attacks. Precisely, our attack contains 2$N$ samples (instead of 1$N$); with customized weights $1/N - 1/\kappa N$ and $1/\kappa N$ (instead of uniformly $1/N$) (see equation (10)). In addition, at each iteration, our attack considers different candidates and select the optimal one, which takes into account the cluster information of the dataset, see Figure 2b.
 - ReLU activation is **not** a requirement of our attack method and in fact, many of architectures in our numerical experiments  used SiLU activations. Nevertheless, we have to treat ReLU and other smooth activation separately in our theoretical analysis since ReLU is not-differentiable.

## 2. Additional numerical experiments and details of computational parameters

- To address the request on ImageNet dataset, in the revised manuscript, we have added these experiments using the same settings as in RobustBench and amended to Table 1. Similar to our existing experiments on CIFAR-10 and CIFAR-100, WDA($\kappa=1$) achieves robust accuracy lower than APGD and comparable to ensemble-based methods AA and $A^3$. Furthermore, WDA ($\kappa=2$) also produces lower robust accuracy values compare to WDA ($\kappa=1$).
- To address the request on scalability of our algorithm, in the revised manuscript, we have added these numerical results in Figure 3c. We report the runtimes (in seconds) of our WDA and $A^3$ on ImageNet dataset with number of sample ranging from 1000 to 5000 and observe that WDA has lower computational time and grows slower than $A^3$, verifying the scalability of our method.
- To address the questions related to step size, we clarify that from Section 6.2 (Ablation Study), for all remaining experiment, we fix the step size to be 0.2 in which involving $r_2=0.5$ and  $\alpha\in\\{0.01, 0.02\\}$ in which involving $r_{\infty}\in \\{4/255, 8/255\\}$.

## 3. Improvements on the clarity of the manuscript

We corrected the typos, added preliminaries on WDRO and rephrased several parts for readability. These changes are highlighted in the revised manuscript.


*We thank the reviewers again for taking their time and effort to provide us with valuable feedback, and we are eager to address any further concerns from the reviewers.*

---

> ### Author Response · Authors · 2025-11-26
> **A gentle reminder**
>
> Dear Reviewers,
>
> Once again, we want to thank you for your thoughtful comments. As the discussion period will end by the next week, could you please let us know whether our response addresses your concerns? We are happy to provide any additional clarifications or results to resolve your doubts.
>
> Best regards,
>
> The authors.

---

### Meta-Review · Area_Chair_mbSF · 2026-01-06

**Summary:**

This paper studies Wasserstein distributionally robust optimization for deep networks, proposing tighter Lipschitz-based certificates and a Wasserstein Distributional Attack (WDA) that constructs adversarial distributions rather than point-wise perturbations. The authors provide theoretical bounds for ReLU and smooth networks and empirical evaluations on standard benchmarks.

The reviewers agree the paper is technically solid, with careful theory and extensive experiments, but raised consistent concerns. In particular, the novelty over prior work on local Lipschitz certificates and distributional robustness is limited, the assumptions required for tightness are restrictive, and the practical impact of the theoretical results is unclear. Reviewers also questioned whether WDA represents a fundamentally new threat model or a reweighting of existing attacks, and noted gaps in clarity, motivation, and positioning relative to strong baselines.

The author response clarified several points and added experiments, but did not fully address the core concerns about conceptual novelty and significance. Taking the reviews, discussion, and response together, the contribution does not meet the bar for acceptance at this venue.

**Reviewer Concerns:**

Please see my summary.

**Reviewer Scores:**

It is difficult to say.  Overall, the authors provided some solid rebuttal, but it's a subjective judgement for the reviewer whether they would like to raise their score.

---

### Decision · Program_Chairs · 2026-01-26

Reject